# The Impact and Vaccination Coverage of Seasonal Influenza among Children Aged 6–59 Months in China in 2017–2018: An Internet Panel Survey

**DOI:** 10.3390/vaccines10040630

**Published:** 2022-04-18

**Authors:** Hangjie Zhang, Xiang Ren, Keqing Tian, Jianxing Yu, Aiqing Zhu, Lijie Zhang, George Fu Gao, Zhongjie Li

**Affiliations:** 1NHC Key Laboratory of Biosafety, National Institute for Viral Disease Control and Prevention, Chinese Center for Disease Control and Prevention, Beijing 100101, China; hjzhang@cdc.zj.cn; 2The Center for Disease Control and Prevention of Zhejiang Province, Hangzhou 310051, China; 3Chinese Field Epidemiology Training Program, Chinese Center for Disease Control and Prevention, Beijing 100101, China; tiankq1989@163.com (K.T.); zhanglj@chinacdc.cn (L.Z.); 4Key Laboratory of Surveillance and Early-Warning on Infectious Disease, Division of Infectious Disease, Chinese Center for Disease Control and Prevention, Beijing 100101, China; renxiang@chinacdc.cn (X.R.); yujianxing@icdc.cn (J.Y.); zhuaq@chinacdc.cn (A.Z.)

**Keywords:** ILI, children, tertiary hospital, influenza vaccine, willingness, internet survey

## Abstract

Seasonal influenza vaccination is highly recommended for 6–59-month-old children. To determine the impact of seasonal influenza and the factors affecting influenza vaccine uptake among children, we conducted an opt-in Internet panel survey of parents from 21 March 2018 to 1 April 2018. Overall, 40.5% (1913/4719) of children experienced influenza-like illness (ILI), 92.4% of parents sought medical care for children with ILI (outpatients: 61.2%, inpatients: 12.8%), 39.6% of parents preferred to take their sick child to a tertiary hospital, and 57.3% of family members requested leave to care for children with ILI. There was a median of three days of absenteeism (2, 5) per sick child, and 39.4% of children received the influenza vaccine during the 2017–2018 influenza season. Vaccine coverage among children aged 6–11 months and 48–59 months was lower than that among 12–47-month-old children. The top three reasons for not vaccinating were: the influenza vaccine was not recommended by healthcare workers (21.1%), no knowledge about the influenza vaccine (19.2%), and lack of confidence in the vaccine’s effectiveness (14.3%). Our findings highlight the need for awareness about the severity of influenza, hygiene behavior, and effectiveness of the influenza vaccine among children and their family members in China.

## 1. Introduction

Seasonal influenza infection circulates worldwide and causes a high rate of morbidity and mortality annually [1,2,3,4]. Young children have a higher annual incidence of influenza, and a higher rate of associated complications compared with adults [5]. The estimated overall attack rate in China was reported to be approximately 5.5% in all age groups, with the highest attack rate observed in 0–4-year-old children (31.9%) [6]. Influenza is a serious global healthcare issue, and substantial challenges are involved in primary care and hospital services for children. Parents requiring time off work to provide care for infected children, and subsequent secondary transmission in households, cause a substantial health burden in terms of healthcare costs and indirect economic losses, such as lost productivity [7].

Parental knowledge about influenza, attitudes, and prevention practices for influenza in children are essential factors for preventing influenza. Additionally, influenza vaccination (IV) has been shown to have consistently high levels of efficacy in reducing the risk of influenza and related complications among children [8]. High-risk groups (e.g., older people, children aged between 6 and 59 months, healthcare workers, pregnant women, and patients with chronic diseases) are prioritized for annual IV [9,10]. However, IV coverage in China has been reported to be relatively poor, with national IV coverage of 1.5% to 2.2% between 2004 and 2014 [11], and average coverage among children aged 5 or below of 26% [12]. Low uptake of IV also contributes to the burden of influenza disease [13].

Increasing uptake of IV is an important issue to be addressed worldwide [14,15,16]. To increase IV coverage, it is important to understand the factors affecting public acceptance and the barriers against IV. Multiple studies have evaluated knowledge about the efficacy and safety of the vaccine, which has been found to be an important factor in the uptake of IV across different age groups [17,18,19,20]. Healthcare professionals’ recommendations are another important factor increasing the acceptance of the vaccine [18,19,21]. Concerns about side effects [18,19] and perceived low susceptibility of contracting influenza [22,23] have also been reported as reasons for not receiving vaccines.

Parents are the primary decision-makers regarding childhood vaccinations including IV; thus, increasing parents’ awareness and willingness may help improve rates of pediatric influenza vaccination. Research about vaccination coverage and factors affecting the decision to vaccinate are only a small number [24,25,26,27]. These studies focused on specific populations, such as children with chronic disease, or children aged >6 months who had been taken to an emergency care center. Another study included parents from a single elementary school in Utah. The findings of these studies suggested that parents are motivated to vaccinate their children against influenza by desire to prevent influenza, a physician’s recommendation, or discussion.

In order to understand the current status and awareness of IV to promote IV uptake among children aged 6–59 months in China, Internet surveys provide a new investigation tool with the advantages of relative cost-effectiveness, greater speed of data collection, and the ability to obtain a large number of respondents in subgroups of interest [28]. In previous studies, we used television and Internet survey methods for examining IV in the general population and among nurses in China [11,29]. Therefore, in the current study we used an Internet panel survey (mobile app “Small Bean Vaccine” platform) to investigate the impact of influenza outbreaks on families, including burden, parental perception, and patterns of action toward childhood influenza, as well as parental beliefs and barriers to administering the influenza vaccine to children in the 2017–2018 influenza season. The current study was intended to shed light on parents’ decision-making processes, and to provide data to guide the development of recommendations for increasing seasonal IV rates in children.

## 2. Materials and Methods

### 2.1. Study Design

From 22 March to 1 April 2018, we conducted an opt-in Internet survey in six provinces, including Henan, Shandong, and Tianjin in northern China, and Guangdong, Guangxi, and Sichuan in southern China. Provinces were selected to represent the range of socioeconomic development and geographical climate zones across China. We used a mobile app called the “Small Bean Vaccine” platform (Shenzhen Threegene Technology Co., Ltd., Shenzhen, China, http://www.yeemiao.com/, accessed on 5 April 2022, which has 10 million family users, accounts for approximately 10% of ≤5-year-old children, and covers nearly 32 provinces in China. Our study panel consisted of registered parents of children who provided basic information to the platform. Invitations containing URL links to the Internet survey were sent to a subset of 10% of users as randomly selected panelists, via a text messaging service using the “Small Bean Vaccine” mobile app in each of the six provinces. Assuming an influenza-like illness (ILI) incidence rate of 30% [6], with a precision level of 1%, a 95% confidence level and a design effect of two, we planned to collect 4000 valid surveys for all of the six provinces combined.

### 2.2. Questionnaire and Data Collection

We used an electronic questionnaire tool to design the online questionnaire and collect data from panelists. The questionnaire included four components: (i) socio-demographic characteristics of children, including province, age, and sex; (ii) self-reported episodes of ILI, defined as reported fever or body temperature ≥ 38 °C, and cough or sore throat, experienced from 1 October 2017 to 1 March 2018, healthcare-seeking behavior and, for children with ILI, days of absence among family members caused by ILI; and (iii) self-reported IV in the 2017/2018 influenza season, and the main reason for receiving or not receiving vaccination from a list of possible reasons. We used screening questions to confirm the eligibility of visitors to the website. Respondents who did not live within one of the six study provinces or who were not 6–59 months old were excluded. Respondents with high levels of missing data (≥50% of items missing), who completed the survey in less than 30 s, or who gave the same answer to 10 consecutive items, were also excluded from the study [28,30]. The study protocol and questionnaire were approved by the ethical review committee at the Chinese Center for Disease Control and Prevention (China CDC, Beijing, China).

### 2.3. Statistical Analysis

To compare demographic and epidemiological characteristics, we used chi-square tests or Fisher’s exact tests for categorical variables, and Wilcoxon rank-sum or Kruskal–Wallis tests for continuous variables, as appropriate. Statistical analysis was performed with Excel software (Microsoft Co., USA) and SPSS (v18.0, SPSS, Chicago, IL, USA). Two-sided *p*-values of <0.05 were considered statistically significant.

## 3. Results

### 3.1. Characteristics of Study Population and Estimation of Influenza-Associated Illness

From 22 March to 1 April 2018, we sent 40,420 invitations to parents of children aged under 5 years, registered in the app platform in the six provinces. Of these, 7323 (18.1%) visited the website and completed the survey. After excluding 2604 (35.6%) surveys that did not meet the data completeness or inclusion criteria, a total of 4719 valid surveys were included in our final analysis. Among these enrolled individuals, 1913 respondents self-reported suffering from ILI, with a winter prevalence of 40.5% (1913/4719) (Table 1). The ILI rate was significantly different in different age groups (*p* < 0.001): children aged 48–59 months (64.0%) had a significantly higher rate than the overall ILI rate, whereas those aged 6–11 months (29.9%) exhibited a rate that was significantly lower than overall ILI rate. The ILI rate was not significantly different between cities in northern and southern China (*p* = 0.152). The rates of self-reported ILI in Guangxi, Sichuan, and Tianjin were greater than the overall average ILI rate (*p* < 0.001).

### 3.2. IV Coverage among Children Aged 6–59 Months in China in 2017–2018

Despite the many previous studies demonstrating the importance of IV, rates of IV remain lower than those for other pediatric vaccines [31]. In this study, only 39.4% (1857/4719) of children were reported to have received the influenza vaccine during the 2017–2018 season. In addition, vaccination among children aged 24–35 months had significantly higher coverage (51.1%) compared with that in other age groups, whereas children aged 6–11 months had the lowest rate of coverage (25.9%) (*p* < 0.001). Interestingly, vaccine coverage in southern China was much higher than that in northern China (45.7% vs. 29.6%) (*p* < 0.001). Children in Guangdong had the highest vaccination coverage (47.0%), followed by Guangxi (45.9%), Sichuan (43.2%), Henan (35.2%), Tianjin (25.7%), and Shandong (21.5%) (Table 2).

### 3.3. Knowledge and Attitudes toward Influenza Infection and Prevention

Among the 4719 enrolled individuals, 78.4% (3700/4719) reported knowing that influenza is different from a common cold, and 81.8% (3861/4719) reported knowing that influenza can cause severe consequences such as hospitalization, severe complications, and death. Parents in Tianjin were the most likely to distinguish influenza and the common cold (86.3%), and severity of the flu (88.9%). There were no significant differences in these factors between the parents of children in different age groups (*p* = 0.572 and 0.975).

It should be noted that the children of parents with knowledge of differences between influenza and the common cold were less likely to suffer ILI compared with those without this knowledge (*p* < 0.001). However, there was no significant difference between parents with and without awareness of the potentially severe consequences of influenza (*p* = 0.541). Prevention awareness, including hand-washing, mask-wearing, and self-segregation, differ between age groups and provinces. The overall self-reported hand-washing rate was found to be 66.1%. The self-reported mask-wearing rate and the self-segregation rate were 40.0% and 83.7%, respectively (Table 3). Tianjin was found to have the highest prevention awareness rates, including washing hands (71.1%) and mask-wearing (52.4%) (*p* < 0.001). Parents with children aged 48–59 months had the highest rate of awareness of hand washing (62.7%) and mask-wearing (45.7%) (*p* = 0.009). Importantly, hygiene behaviors such as frequent hand-washing (*p* = 0.005) and mask-wearing (*p* < 0.001) were more effective for preventing children contracting influenza, although the rate of self-segregation was not significantly different (*p* = 0.245) (Table 4).

### 3.4. Healthcare-Seeking Behaviors Related to Influenza Infection and the Impact of Children Suffering from ILI on the Family

We examined information regarding healthcare-seeking behaviors among individuals targeted in the Internet survey who were reported to have ILI symptoms (Table 5). It should be noted that the parents in our sample had high rates of healthcare-seeking awareness and behaviors for their children with ILI symptoms (92.4%). The healthcare-seeking rates for parents of 36–47-month-old and 48–59-month-old children were higher than those of parents of children in other age groups (*p* < 0.001), which may be related to the ILI attack rate. Parents in the provinces of Henan, Sichuan, and Tianjin had higher healthcare-seeking rates than average, whereas patents in Shandong had the lowest healthcare-seeking rate (88.8%). Parents’ first choice of medical treatment for their children was outpatient/emergency care (61.2%), followed by drugstore (22.0%), inpatient care (12.8%), and self-treatment (3.8%). Compared with other regions, parents in Guangdong province were found to have the highest frequency of seeking medicine from outpatient/emergency care (71.5%). For children with ILI, more parents preferred to take sick children to tertiary hospitals (30.9%), followed by secondary hospitals (21.2%), clinic/village doctors (20.0%), and community health service centers (18.7%), particularly for 48–59-month-old children. In addition, parents in Tianjin preferred tertiary hospitals for their first visit. However, parents in Henan preferred visiting private clinic/village doctors rather than municipal hospitals.

During seasonal influenza epidemics, high attack rates of generally mild and self-limiting illnesses cause a considerable burden for outpatient medical visits, and a small fraction of infections are severe, requiring hospitalization [32]. The median medical cost for a family in 2017/2018 influenza season in China was 300–799 Yuan. Moreover, 71.6% of families self-paid their medical fee completely, with particularly high rates in Shandong (81.1%) and Tianjin (78.1%). To care for children with ILI, 57.3% of family members asked for leave, taking a median of 3 (2, 5) days of absenteeism per sick child. Families of 48–59-month-old children exhibited the highest likelihood of taking leave to care for their children, and the median number of days of absenteeism was 5 (3, 10) days (Table 6). Interestingly, a higher proportion of families in northern provinces required leave (61.7%), particularly in Shandong (70.4%) and Tianjin (71.0%) province, with a median of 4.5 (2, 6) days of absenteeism.

### 3.5. Factors Related to Willingness to Be Vaccinated for Influenza Vaccine

The willingness of parents for their children to have IV in the next influenza season was found to be 88.6%. Individuals in southern provinces were found to be more likely to be vaccinated against influenza than those in northern provinces (*p* < 0.001). Guangdong had the largest proportion of individuals with the intention to vaccinate their children against influenza for the 2018/2019 influenza season. The gender of children made no significant difference to vaccination intention rates (*p* = 0.500). However, children in younger age groups were more likely to be vaccinated against influenza, and the parents of children aged 6–11 months had the highest rate of vaccination willingness (95.0%), followed by parents of children aged 12–23 months (88.2%), 24–35 months (87.9%), 36–47 months (86.8%), and 48–59 months (82.2%) (*p* = 0.002) (Table 7).

In addition, 47.2% of respondents who self-reported the intention for their children to receive IV believed that influenza vaccines provide protection for their children, while 21.6% of respondents self-reported an intention for IV because the influenza vaccine was recommended by their doctors, and 11.0% of respondents self-reported an intention to receive IV on the condition that the influenza vaccines are free. Recommendation from doctors and knowledge about the influenza vaccine were found to be the most effective factors for increasing the intention for IV. Overall, 21.1% of respondents self-reported that their children would not receive IV because their doctors did not recommend it, whereas 19.2% had not heard about the influenza vaccine, and 14.3% believed that the influenza vaccine does not offer effective protection. In addition, the cost and convenience of vaccination were also found to be important factors (Figure 1).

## 4. Discussion

In the 2017–2018 influenza season, countries in the northern hemisphere, including China, experienced a large number of local outbreaks and severe epidemics [33,34,35]. The influenza virus can exhibit rapid transmission in the community because of high viremic titers and long periods of viral shedding in high-risk groups, such as children and older people [36]. The current results revealed that the overall influenza attack rate was 40.5% among children aged 6–59 months in the 2017/2018 influenza season, which is close to the upper limit for the incidence of ILI during a given season (approximately 15–45% of children). Interestingly, ILIs were significantly more common in children aged 48–59 months (64.0%) compared with those aged 6–11 months (29.9%). Feng et al. have mentioned that influenza virus infection was more frequent in children (2–4 years) than (6–11 months) from the active surveillance for hospitalized ALRI patients in China [37]. The high attack rates in older children may be associated with higher more social activities and increased exposure risk to the influenza virus.

A previous meta-analysis indicated that the overall healthcare-seeking rate of individuals with ILI is 52% [38]. However, in a higher-pandemic period such as the 2009 H1N1 pandemic, the healthcare-seeking rate was 74% in individuals [39,40]. Importantly, children had a higher rate of healthcare-seeking than adults [41,42]. Understanding of the proportions and pathways of healthcare-seeking behavior of parents is important for evaluating the actual burden and fatality rate of seasonal influenza in children. The current results revealed that the healthcare-seeking rate of children with ILI symptoms was 92.4% during the 2017/2018 influenza season. Outpatient/emergency care is the first choice of healthcare of parents, followed by drugstores. Some children were hospitalized with serious illness. For children with ILI, the proportion of parents selecting tertiary and secondary hospitals as their first choice was significantly higher than the proportion selecting clinic/village doctors and community health service centers as their first choice. In addition, regional economic development may influence the healthcare choices of parents. Parents in Tianjin preferred tertiary hospitals, whereas those in Henan preferred private clinic/village doctors rather than municipal hospitals.

Seasonal influenza epidemics can cause a significant economic burden and severe impacts on family life. Understanding the regional and population characteristics of the influenza-associated burden is crucial for guiding policy-making for IV. It has been reported that the cost per capita in the United States and European countries ranges from $27 to $63, which is higher than that in middle-income Asian and African countries [43]. One previous study reported that the average total cost per episode of influenza in a child (aged 0–59 months) in an outpatient setting was 768.0 Yuan (95% CI: 686.8–849.3) in Suzhou, China in 2011–2017 [44]. Similarly, the estimated median direct-cost of influenza-associated illness of a family was found to be 300–799 Yuan in our study. Although the influenza epidemic in 2017/2018 was serious, the incurred cost for most families did not significantly increase. Moreover, 71.6% of families were required to self-pay the entire medical fee. The basic medical system providing medical insurance for children in China is developing slowly, and there is only partial coverage of finance-reimbursed IV in some regions and some populations, such as school children [45]. Therefore, considering commercial guarantees to reduce family risk is recommended. In addition to direct costs of medication use and healthcare services, indirect costs must be considered, such as absenteeism for both the child and their parents [46]. The current results revealed that 57.3% of parents requested leave, with a median duration of 3 (2, 5) days of absenteeism per sick child.

Health literacy influences the health behavior and services of individuals and family members. A previous study in Hong Kong reported that parents’ health knowledge and access to health information concerning seasonal influenza was related to seasonal influenza prevention [47]. Of the parents included in our survey, approximately 80% had knowledge about the severity of influenza. Importantly, as shown in our data, better knowledge about influenza in the family was associated with a lower rate of ILI among children. A previous study found that health-protective behaviors, such as receiving a vaccine, mask-wearing, and frequent hand-washing, can reduce the risk of influenza transmission [48]. In the 2017/2018 influenza season, our data indicated that rates of hand-washing, mask-wearing and self-segregation in children aged 6–59 months children were relatively high, particularly for 48–59-month-olds. Moreover, frequent hand-washing and mask-wearing were more important and effective hygiene behaviors for contracting influenza. It is important to provide influenza-related information to enhance parents’ health literacy in relation to their children, and to support the development of healthy behaviors.

The influenza vaccine is the most effective public health measures for preventing influenza. The childhood vaccination rate in Hong Kong is reported to be 37.9% among those aged 12 or below) [17]. However, the childhood vaccination rate in developed countries is relatively high, with coverage of 57.9% among children aged 6 months to 17 years in the United States, and 64% among children aged 6–13 years in Japan, respectively [17,21]. In this study, only 39.4% of children were reported to have received the influenza vaccine during the 2017–2018 season. In addition, 24–35-month-olds had significantly higher vaccination coverage (51.1%), while 6–11-month-olds had the lowest coverage (25.9%). The vaccine coverage in southern provinces was substantially higher than that in northern provinces (45.7% vs. 29.6%). Though there was no difference of ILI attack rate among children in north or south China in 2017–2018, the vaccine coverage in southern areas was much higher. Factors including economic development level or knowledge to influenza vaccine would likely influence the influenza vaccine uptake in the south or north. Moreover, parents having more willingness to vaccinate their children for the next influenza season in the south also confirmed this phenomenon.

In addition, 47.2% of respondents who self-reported the intention of receiving IV believe that the influenza vaccines have protective effects for their children, while 21.6% of respondents self-reported the intention for IV because the influenza vaccine was recommended by their doctors, and 11.0% of respondents self-reported the intention of receiving an IV on the condition that the influenza vaccines are free. Recommendation from one’s doctor and knowledge of influenza vaccine were found to be the most effective factors for increasing the intention for IV. Overall. 21.1% of respondents self-reported the intention to not receive IV because their doctors did not recommend it, while 19.2% had not heard about the influenza vaccine, and 14.3% believed that the influenza vaccine offered no protection. In addition, the cost of the influenza vaccine and the convenience of receiving the vaccine were also important factors.

Several limitations of this study should be noted. First, Internet panel surveys typically have low recruitment participation rates with a low likelihood of potential respondents opting in, despite the advantage of having a known denominator (sampling frame). Second, because of sample selection bias, most of our respondents were insured, potentially exhibiting different patterns of vaccination and healthcare utilization compared with uninsured individuals. Meanwhile, the Internet panel survey method may have increased the likelihood of including participants who lived in urban areas compared with participants who lived in rural areas. Third, our estimates of cases averted only included the direct effects of the vaccine among vaccinated individuals, and did not account for indirect effects caused by reduced transmission of influenza. Thus, the true impact of IV may be greater than that estimated in this study.

## 5. Conclusions

Our study provides valuable insight into the level of awareness about influenza illness and the insufficient IV childhood coverage rate in a Chinese population. Vaccination willingness was positively associated with knowledge about the influenza vaccine, recommendations from doctors, and free vaccination. This insight has important implications for the development of vaccination policy by local governments regarding IV.

## Figures and Tables

**Figure 1 vaccines-10-00630-f001:**
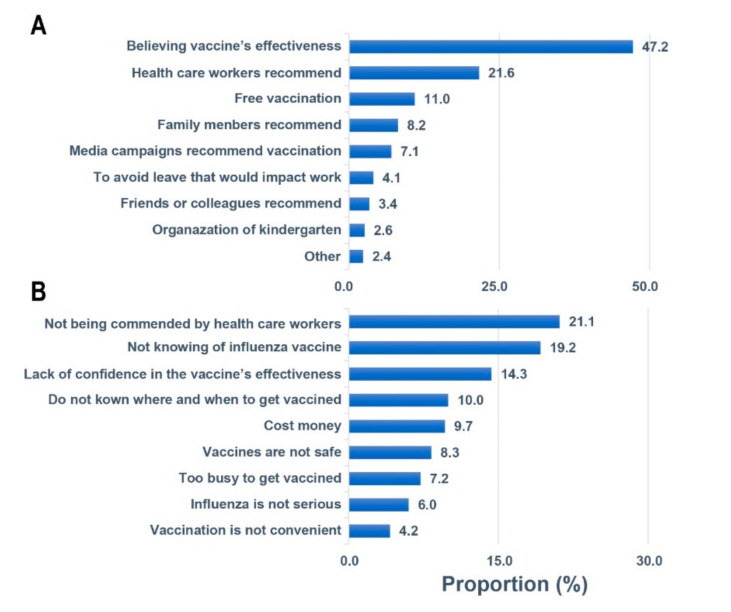
Reasons for children receiving or not receiving the influenza vaccine among parents in China from 2017 to 2018. (**A**) Main reason for receiving the influenza vaccine among 1857 vaccinated children. (**B**) Main reason for not receiving the influenza vaccine among 2394 unvaccinated children.

**Table 1 vaccines-10-00630-t001:** Sociodemographic characteristics of participants.

Characteristics	Population *n* = 4719 (%)	ILI Symptoms *n* = 1913 (%)	*p*-Value
Sex			0.174
Male	2471 (52.4)	1041 (42.1)	
Female	2248 (47.6)	872 (38.8)	
Age in months			<0.001
6–11	1868 (39.6)	559 (29.9)	
12–23	1604 (34.0)	735 (45.8)	
24–35	620 (13.1)	272 (43.9)	
36–47	402 (8.5)	203 (50.5)	
48–59	225 (4.8)	144 (64.0)	
Region			0.152
North	1854 (39.3)	728 (39.2)	
South	2865 (60.7)	1185 (41.3)	
Province			<0.001
Guangdong	1303 (27.6)	479 (36.8)	
Henan	846 (17.9)	305 (36.1)	
Guangxi	994 (21.1)	456 (45.9)	
Sichuan	545 (11.5)	250 (45.9)	
Shandong	716 (15.2)	287 (40.1)	
Tianjin	315 (6.7)	136 (43.2)	

**Table 2 vaccines-10-00630-t002:** Proportion of children who received influenza vaccination.

Characteristics	Number Vaccinated	Number Unvaccinated	Unknown	*p*-Value
Sex				0.958
Male	972 (39.3)	1251 (50.6)	248 (10.0)	
Female	885 (39.4)	1143 (50.8)	220 (9.8)	
Age in months				<0.001
6–11	551 (25.9)	1104 (59.1)	213 (11.4)	
12–23	733 (45.7)	726 (45.3)	145 (9.0)	
24–35	319 (51.5)	243 (39.2)	58 (9.4)	
36–47	179 (44.5	189 (47.0)	34 (8.5)	
48–59	75 (33.3)	132 (58.7)	18 (8.0)	
Region				
North	548 (29.6)	1109 (59.8)	197 (10.6)	<0.001
South	1309 (45.7)	1285 (44.9)	271 (9.5)	
Province				<0.001
Guangdong	612 (47.0)	594 (45.6)	97 (7.4)	
Henan	350 (35.2)	516 (51.9)	128 (12.9)	
Guangxi	388 (45.9)	347 (41.0)	111 (13.1)	
Sichuan	309 (43.2)	344 (48.0)	63 (8.8)	
Shandong	117 (21.5)	383 (70.3)	45 (8.3)	
Tianjin	81 (25.7)	210 (66.7)	24 (7.6)	
Total	1857 (39.4)	2394 (50.7)	468 (9.9)	

**Table 3 vaccines-10-00630-t003:** Knowledge of influenza prevention measures among family members.

Characteristics	6–59 Months					*p*-Value	Province						*p*-Value	Total
	6–11	12–23	24–35	36–47	48–59		Guangdong	Henan	Guangxi	Sichuan	Shandong	Tianjin		
Distinction of flu and cold						0.572							<0.001	
Yes	1457 (78.0)	1251 (78.0)	498 (80.3)	319 (79.4)	175 (77.8)		992 (76.1)	767 (77.2)	633 (74.8)	547 (80.2)	462 (84.8)	272 (86.3)		3700 (78.4)
No	63 (3.4)	58 (3.6)	22 (3.5)	19 (4.7)	12 (5.3)		49 (3.8)	37 (3.7)	30 (3.5)	34 (4.7)	15 (2.8)	9 (2.9)		174 (3.7)
Unknown	348 (18.6)	295 (18.4)	100 (16.1)	64 (15.9)	38 (16.9)		262 (20.1)	190 (19.1)	183 (21.6)	108 (15.1)	68 (12.5)	34 (10.8)		845 (17.9)
Serious illness induced by flu						0.975							<0.001	
Yes	1520 (81.4)	1319 (82.2)	509 (82.1)	334 (83.1)	179 (79.6)		1075 (82.5)	747 (75.2)	680 (80.4)	612 (85.5)	467 (85.7)	280 (88.9)		3861 (81.8)
No	73 (3.9)	57 (3.6)	21 (3.4)	14 (3.5)	8 (3.6)		39 (3.0)	52 (5.2)	32 (3.8)	19 (2.7)	17 (3.1)	14 (4.4)		173 (3.7)
Unknown	275 (14.7)	228 (14.2)	90 (14.5)	54 (13.4)	38 (16.9)		189 (14.5)	195 (19.6)	134 (15.8)	85 (11.9)	61 (11.2)	21 (6.7)		685 (14.5)
Washing hands						0.009							<0.001	
Yes	1172 (62.7)	1103 (68.8)	423 (68.2)	275 (68.4)	146 (64.9)		906 (69.5)	596 (60.0)	578 (68.3)	425 (59.4)	390 (71.6)	224 (71.1)		3119 (66.1)
No	557 (29.8)	403 (25.1)	156 (25.2)	108 (18.4)	59 (26.2)		333 (25.6)	311 (31.3)	198 (23.4)	230 (32.1)	135 (24.8)	76 (24.1)		1283 (27.2)
Unknown	139 (7.4)	98 (6.1)	41 (6.6)	19 (4.7)	20 (8.9)		64 (4.9)	87 (8.8)	70 (8.3)	61 (8.5)	20 (3.7)	15 (4.8)		317 (6.7)
Mask wearing						<0.001							<0.001	
Yes	854 (45.7)	566 (35.3)	217 (35.0)	155 (38.6)	94 (41.8)		478 (36.7)	379 (38.1)	293 (34.6)	297 (41.5)	274 (50.3)	165 (52.4)		1886 (40.0)
No	917 (49.1)	957 (59.7)	374 (60.3)	235 (58.5)	125 (55.6)		774 (59.4)	551 (55.4)	479 (58.7)	391 (54.6)	253 (46.4)	142 (45.1)		2608 (55.3)
Unknown	97 (5.2)	81 (5.0)	29 (4.7)	12 (3.0)	6 (2.7)		51 (3.9)	64 (6.4)	56 (6.6)	28 (3.9)	18 (3.3)	8 (2.5)		225 (4.8)
Self-segregation						0.093							0.011	
Yes	1568 (83.9)	1362 (84.9)	512 (82.6)	330 (82.1)	180 (80.0)		1127 (86.5)	793 (79.8)	704 (83.2)	596 (83.2)	463 (85.0)	269 (85.4)		3952 (83.7)
No	236 (12.6)	189 (11.8)	85 (13.7)	66 (16.4)	34 (15.1)		141 (10.8)	157 (15.8)	109 (12.9)	101 (14.1)	65 (11.9)	37 (11.7)		610 (12.9)
Unknown	64 (3.4)	53 (3.3)	23 (3.7)	6 (1.5)	11 (4.9)		35 (2.7)	44 (4.4)	33 (3.9)	19 (2.7)	17 (3.1)	9 (2.9)		157 (3.3)

**Table 4 vaccines-10-00630-t004:** Influence of knowledge about influenza prevention measures on influenza infection.

Characteristics	Influenza-Positive *n* = 1913 (%)	Influenza-Negative *n* = 2806 (%)	*p-Value*
Distinction of flu and cold			<0.001
Yes	1458 (76.2)	2242 (79.9)	
No	96 (5.0)	78 (2.8)	
Unknown	359 (18.8)	486 (17.3)	
Serious illness induced by flu			0.541
Yes	1518 (79.4)	2343 (83.5)	
No	64 (3.3)	109 (3.9)	
Unknown	331 (17.3)	354 (12.6)	
Washing hands			0.005
Yes	1247 (65.2)	1872 (66.7)	
No	572 (29.9)	711 (25.3)	
Unknown	94 (4.9)	223 (7.9)	
Mask wearing			<0.001
Yes	647 (33.8)	1239 (44.2)	
No	1208 (63.1)	1400 (49.9)	
Unknown	58 (3.0)	167 (6.0)	
Self-segregation			0.245
Yes	1612 (84.3)	2340 (83.4)	
No	264 (13.8)	346 (12.3)	
Unknown	37 (1.9)	120 (4.3)	

**Table 5 vaccines-10-00630-t005:** Medical treatment-seeking and medical expenses.

Characteristics	6–59 Months					Province						Total
	6–11	12–23	24–35	36–47	48–59	Guangdong	Henan	Guangxi	Sichuan	Shandong	Tianjin	
Medical treatment seeking												
Yes	516 (92.3)	677 (92.1)	247 (90.8)	191 (94.1)	136 (94.4)	434 (90.6)	436 (95.6)	276 (90.5)	271 (94.4)	222 (88.8)	128 (94.1)	1767 (92.4)
No	42 (7.5)	49 (6.7)	25 (9.2)	10 (4.9)	6 (4.2)	43 (9.0)	18 (3.9)	23 (7.5)	15 (5.2)	25 (10.0)	8 (5.9)	132 (6.9)
Unknown	1 (0.2)	9 (1.2)	0 (0)	2 (1)	2 (1.4)	2 (0.4)	2 (0.4)	6 (2.0)	1 (0.3)	3 (1.2)	0 (0)	14 (0.7)
Treatment method												
Outpatient/emergency	415 (58.2)	592 (63.4)	212 (61.8)	162 (59.3)	119 (64.0)	393 (71.5)	356 (59.8)	222 (55.9)	221 (57.3)	188 (58.9)	120 (61.9)	1500 (61.2)
Be hospitalized	111 (15.6)	112 (12.0)	34 (9.9)	31 (11.4)	26 (14.0)	52 (9.5)	93 (15.4)	52 (13.1)	75 (19.4)	28 (8.8)	14 (7.2)	314 (12.8)
Drugstore	158 (22.2)	196 (21.0)	84 (24.5)	66 (24.2)	36 (19.4)	85 (15.5)	141 (23.3)	99 (24.9)	77 (19.9)	84 (26.3)	54 (27.8)	540 (22.0)
Rest with no treat	29 (4.1)	33 (3.5)	13 (3.8)	14 (5.1)	4 (2.2)	19 (3.5)	14 (2.3)	23 (5.8)	13 (3.4)	18 (5.6)	6 (3.1)	93 (3.8)
Unknown	0 (0)	1 (0.1)	0 (0)	0 (0)	1 (0.5)	0 (0)	0 (0)	1 (0.3)	0 (0)	1 (0.3)	0 (0)	2 (0.1)
Hospital choosing												
Tertiary hospitals	203 (30.6)	302 (36.0)	95 (29.7)	88 (35.2)	69 (41.3)	203 (38.2)	114 (19.9)	104 (29.1)	112 (31.8)	139 (51.5)	85 (54.5)	757 (33.8)
Secondary hospitals	165 (24.8)	200 (23.9)	76 (23.8)	46 (18.4)	31 (18.6)	127 (23.9)	124 (21.6)	90 (25.2)	105 (29.8)	40 (14.8)	32 (20.5)	518 (23.1)
Community health service	130 (19.6)	166 (19.8)	74 (23.1)	62 (24.8)	27 (16.2)	139 (26.2)	117 (20.4)	74 (20.7)	66 (18.8)	47 (17.4)	16 (10.3)	459 (20.5)
Clinic/village doctor	160 (24.1)	166 (19.8)	73 (22.8)	53 (21.2)	38 (22.8)	59 (11.1)	215 (37.5)	85 (23.8)	66 (18.8)	42 (15.6)	23 (14.7)	490 (21.9)
Unknown	6 (0.9)	4 (0.5)	2 (0.6)	1 (0.4)	2 (1.2)	3 (0.6)	3 (0.5)	4 (1.1)	3 (0.9)	2 (07)	0 (0)	15 (0.6)
Medical expense (¥)												
0–299	208 (40.5)	224 (33.1)	91 (36.8)	52 (27.2)	28 (20.6)	168 (38.8)	169 (38.8)	113 (40.9)	82 (30.5)	44 (19.8)	27 (21.1)	603 (31.5)
300–799	123 (23.9)	199 (29.4)	77 (31.2)	66 (34.6)	47 (34.6)	147 (33.9)	106 (24.3)	80 (29.0)	75 (27.9)	66 (29.7)	38 (29.7)	512 (26.8)
800–1499	56 (10.9)	109 (16.1)	34 (13.8)	27 (14.1)	20 (14.7)	56 (12.9)	65 (14.9)	29 (10.5)	28 (10.4)	39 (17.6)	29 (22.7)	246 (12.9)
1500–2999	53 (10.3)	71 (10.5)	23 (9.3)	29 (15.2)	20 (14.7)	31 (7.2)	52 (11.9)	20 (7.2)	37 (13.8)	42 (18.9)	14 (10.9)	403 (21.1)
3000	67 (13.0)	61 (9.0)	15 (6.1)	12 (6.3)	21 (15.4)	25 (5.8)	38 (8.7)	25 (9.1)	40 (14.9)	29 (13.1)	19 (14.8)	176 (9.2)
Unknown	7 (1.4)	12 (1.8)	7 (2.8)	5 (2.6)	0 (0)	6 (1.4)	6 (1.4)	9 (3.3)	7 (2.6)	2 (0.9)	1 (0.8)	149 (7.8)
Personal-paid												
100%	367 (71.1)	506 (74.4)	167 (67.6)	136 (71.2)	89 (65.4)	287 (66.1)	315 (72.2)	199 (72.1)	184 (67.9)	180 (81.1)	100 (78.1)	1265 (71.6)
80–99%	31 (6.0)	34 (5.0)	14 (5.7)	11 (5.8)	8 (5.9)	29 (6.7)	17 (3.9)	15 (5.4)	19 (7.0)	9 (4.1)	9 (7.0)	98 (5.5)
50–79%	45 (8.7)	54 (8.0)	28 (11.3)	15 (7.9)	20 (14.7)	22 (5.1)	50 (11.5)	26 (9.4)	41 (15.1)	13 (5.9)	10 (7.8)	162 (9.2)
49–30%	30 (5.8)	26 (3.8)	9 (3.6)	10 (5.2)	7 (5.1)	27 (6.2)	23 (5.3)	11 (4.0)	11 (4.1)	7 (3.2)	3 (2.3)	82 (4.6)
<30%	22 (4.3)	25 (3.7)	11 (4.5)	12 (6.3)	3 (2.2)	43 (9.9)	11 (2.5)	6 (2.2)	3 (1.1)	5 (2.3)	5 (3.9)	73 (4.1)
Unknown	21 (4.1)	32 (4.7)	18 (7.3)	7 (3.7)	9 (6.6)	26 (6.0)	20 (4.6)	19 (6.9)	13 (4.8)	8 (3.6)	1 (0.8)	87 (4.9)

**Table 6 vaccines-10-00630-t006:** Absenteeism among family members of children with ILI.

Characteristics	Absenteeism (%)	Median, Day (IQR *)
Sex		
Male	576 (58.8)	3 (2, 5)
Female	454 (55.6)	3 (2, 5)
Age in months		
6–11	274 (52.2)	2 (1, 2)
12–23	383 (55.3)	4.5 (2.5, 7)
24–35	150 (58.8)	3 (2, 7)
36–47	121 (64.4)	5 (2, 7.5)
48–59	102 (75.0)	5 (3, 10)
Region		
North	489 (61.7)	4.5 (2, 6)
South	541 (53.8)	2 (1, 3)
Province		
Guangdong	263 (58.6)	3 (1.3, 5)
Henan	227 (53.6)	6 (3, 9)
Guangxi	131 (45.5)	4 (2, 7)
Sichuan	147 (54,9)	5 (2, 7)
Shandong	169 (70.4)	5 (2.8, 10)
Tianjin	93 (71.0)	5 (2, 7)
Total	1030 (57.3)	3 (2, 5)

* IQR = interquartile range.

**Table 7 vaccines-10-00630-t007:** Willingness of parents regarding influenza vaccination for their children.

Characteristics	Yes (%)	No (%)	*p*-Value
Sex			0.500
Male	2183 (88.3)	288 (11.7)	
Female	2000 (89.0)	248 (11.0)	
Age in months			0.002
6–11	1690 (90.5)	178 (9.5)	
12–23	1414 (88.2)	190 (11.8)	
24–35	545 (87.9)	75 (12.1)	
36–47	349 (86.8)	53 (13.2)	
48–59	185 (82.2)	40 (17.8)	
Region			<0.001
North	1579 (85.2)	275 (14.8)	
South	2604 (90.9)	261 (9.1)	
Province			<0.001
Guangdong	1202 (92.2)	101 (7.8)	
Henan	878 (88.3)	116 (11.7)	
Guangxi	758 (89.6)	88 (10.4)	
Sichuan	644 (89.9)	72 (10.1)	
Shandong	446 (81.8)	99 (18.2)	
Tianjin	225 (81.0)	60 (19.0)	
Total	4183 (88.6)	536 (11.4)	

## Data Availability

The datasets generated and analyzed during this study are not publicly available due to the institute’s data security and sharing policy, but are available from the corresponding author on reasonable request.

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
