# Peer review of "The Impact and Vaccination Coverage of Seasonal Influenza among Children Aged 6–59 Months in China in 2017–2018: An Internet Panel Survey"

_vaccines, 2022, doi:10.3390/vaccines10040630_

Round 1

Reviewer 1 Report

Zhang et al. provide an excellent review about the level of awareness about influenza ill-ness and the insufficient influenza vaccine childhood coverage rate in a Chinese population. The authors stated that the vaccination willingness was positively associated with knowledge about the influenza vaccine, recommendations from doctors, and free vaccination. This review can be useful for the development of vaccination policy by local governments for influenza vaccination. 

Some of the minor suggestions/modification about the article are as follow;

Please bridge the Gaps related to the literature review in the introduction section.

Describe the p-value wherever the word significant or insignificant is used by the authors especially in the results section.

In the discussion section, higher T cells activation status instead of higher activation status should be mentioned.

The authors have done a good job in using the internet based application for the dissemination and retrieval of survey information regarding their questionnaire. At the same time the authors cannot state anything about chance without any statistical tests. Any close relationship showing strong values should be mentioned in the text where applicable.

However even if the p-values showed significance, such studies have many confounding factors in addition to other fallacies that can only be understood by further epidemiological studies. that could possibly be added to improve the section about limitations of the study. In the current state the study offers an excellent overview about the awareness about the severity of influenza infection, hygiene and influenza vaccine effectiveness in children and merits publication with minor modifications.

Reviewer 2 Report

The manuscript represents a general overview of influenza vaccination in the pediatric population in 6 different regions of China. The group of analyses was divided according to the presentation of symptoms and the use of the vaccine. There are several issues that must be clarified and added 1) the questionnaire itself and its validation, who validated the data, 2) the conditions of the patients that had symptoms, anthropometric measurements, other medical conditions, which could affect the conditions of the patient and their response. 3) there are differences in table 1, amount of individuals in the north and south region were significant and differ in table 6, no significance was stated. However, in table 3, it is clear that the are no differences in absenteeism but parents were willing to vaccinate their children in the south so how do the authors explain this difference?

Table 4 is not complete in the paper. The discussion should be enhancd.

Reviewer 3 Report

The authors present the summary of findings based on an opt-in survey of parents with children that experienced flu-like illness from March-April 2018. The study provides some interesting insights into disease prevalence among different age groups of children that can be correlated with their vaccination status.  It also provides some insight into how parents might make decisions regarding vaccinating their kids, and how regional differences in China might impact such decisions.

The data is presented comprehensively, but in doing so, some key observations are easy to miss and might be confusing to the reader.  For example, while the title highlights in the impact of 'vaccine coverage', statistics regarding vaccine coverage are not discussed until table 6.  It might be helpful to present this data earlier so that a reader can appreciate the other variables in light of vaccine status.  

One unexpected finding, for this reviewer, was that the youngest children in the study had the least % of sick kids (Table I) despite also having the lowest percentage of vaccination (Table 6).  The authors should elaborate on why this relationship might exist -- has it been seen in other similar studies?  The authors provide one rather vague sentence at the end of the first paragraph of the discussion about this point, but it should be expanded on given that most people would likely guess that disease should be most prevalent in the least immunized subset, especially as the youngest kinds might be most vulnerable.  Of note: if this is indeed a valid correlation, it might explain why parents of very young children DO NOT vaccinate their children to the degree seen with older cohorts.

The English could use some more proof reading/editing.  For example, the first sentence of the third paragraph of the introduction is not actually a sentence.  The 4th column in Table 6 should also be changed to "Unknown" from "Unknow".

Finally, a little more information on the  'Small Bean Vaccine" vaccine platform app would be helpful -- I tried to look this up on the internet but could find no information to help truly understand the nature of this app.

Round 2

Reviewer 2 Report

The manuscript was improved and the queries responded in general. I would suggest adding the validation of the questionnaire in the material and methods section. 

Reviewer 3 Report

Thanks to the authors for their revisions.  This is an interesting study and reads far more easily now.